ecology, environmental science, systems biology

zooplankton, food web, trophic niche diversity, metabarcoding, rotifer

**Author for correspondence:**
Andreas Novotny
e-mail: mail@andreasnovotny.se

†Present address: Institute of Marine Research, Bergen, Norway.

# DNA metabarcoding reveals trophic niche diversity of micro and mesozooplankton species

Andreas Novotny, Sara Zamora-Terol† and Monika Winder

Department of Ecology, Environment, and Plant Science, Stockholm University, Svante Arrhenius Väg 20A, 106 91 Stockholm, Sweden

AN, 0000-0001-8910-6183; SZ-T, 0000-0002-7822-3197

Alternative pathways of energy transfer guarantee the functionality and productivity in marine food webs that experience strong seasonality. Nevertheless, the complexity of zooplankton interactions is rarely considered in trophic studies because of the lack of detailed information about feeding interactions in nature. In this study, we used DNA metabarcoding to highlight the diversity of trophic niches in a wide range of micro- and mesozooplankton, including ciliates, rotifers, cladocerans, copepods and their prey, by sequencing *16-* and *18S rRNA* genes. Our study demonstrates that the zooplankton trophic niche partitioning goes beyond both phylogeny and size and reinforces the importance of diversity in resource use for stabilizing food web efficiency by allowing for several different pathways of energy transfer. We further highlight that small, rarely studied zooplankton (rotifers and ciliates) fill an important role in the Baltic Sea pelagic primary production pathways and the potential of ciliates, rotifers and crustaceans in the utilization of filamentous and picocyanobacteria within the pelagic food web. The approach used in this study is a suitable entry point to ecosystem-wide food web modelling considering species-specific resource use of key consumers.

## 1. Introduction

The ability for ecosystems to maintain functionality and productivity under annual and seasonal variation in primary production relies on energy transfer pathways sustained by a network of diverse primary consumers [1,2]. In marine food webs, functionally diverse assemblages of planktonic bacteria, protists and metazoans regulate the flow of energy from primary producers to higher trophic levels [3–5]. While crustacean zooplankton (e.g. copepods and cladocerans) constitute the primary link between phytoplankton and planktivorous fish [6], microzooplankton are main grazers of primary production at times when the biomass of phytoplankton is low or inedible [3,7]. In order to estimate the resilience of marine ecosystems, a mechanistic understanding of resource use by the primary consumers is needed [8]. However, in most food web studies, the trophic niche is based on size or phylogeny due to a lack of detailed information about feeding interactions in nature. Consequently, the entire niche diversity of the zooplankton community is not accurately considered [9,10].

Variation in temporal abundance, feeding traits, size, phenotypic plasticity, growth rate and predation resistance contribute to the total diversity of zooplankton functional groups in marine food webs [11]. While most trophic studies have clustered zooplankton into broad phylogenetic groups [9], recent studies show that incorporating traits, particularly size, has consequences for interpreting food web dynamics and productivity [12–14]. For example, the rotifer phylum contains members of different size classes [15], as well as organisms with various feeding behaviours including filter feeders [16], selective

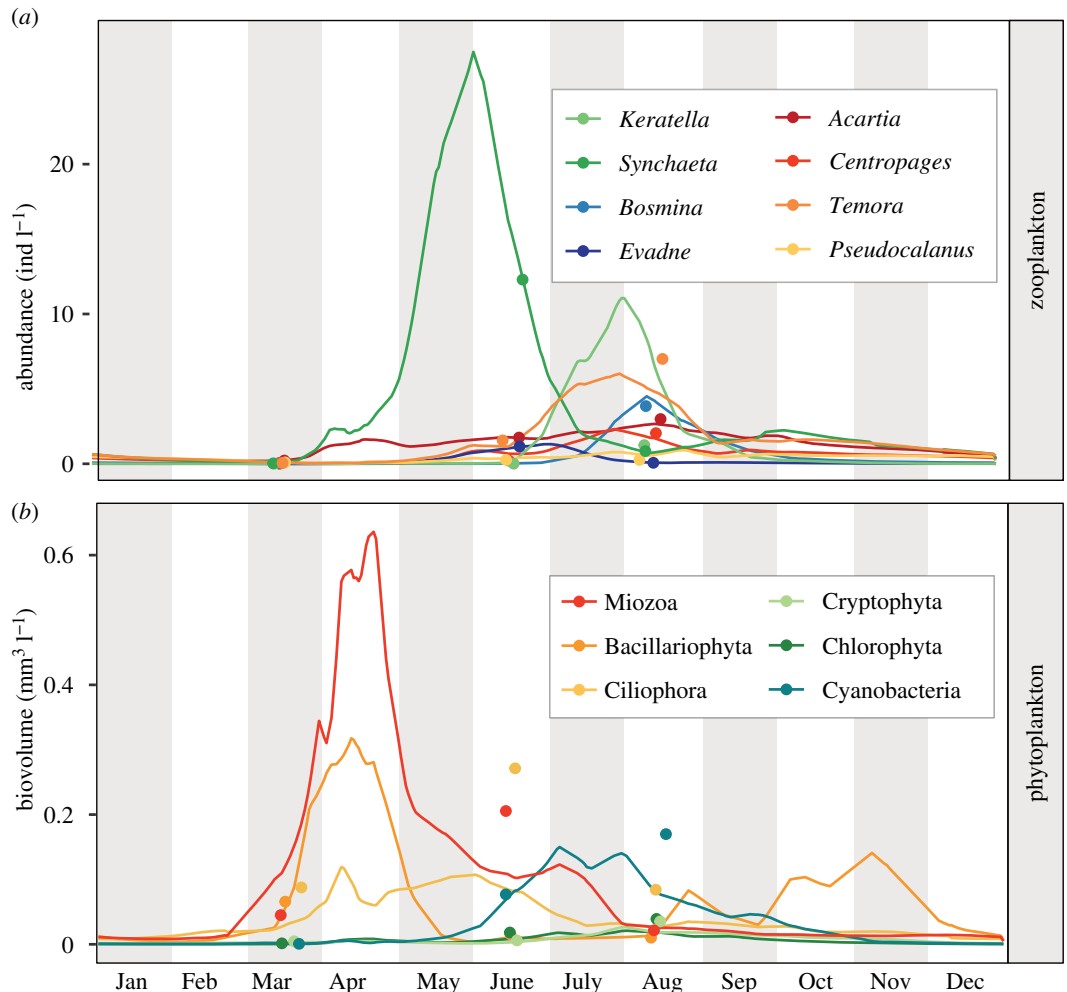

**Figure 1.** (*a*) Abundance of zooplankton and (*b*) biovolume of phytoplankton at Landsort Deep in the Baltic Sea. Interpolated daily means over the years 2006–2018. The data are available at the Swedish national archive for oceanographic data: https://sharkweb.smhi.se/. Samples are taken weekly to bi-weekly during the spring and summer period and monthly during winter. The points indicate (*a*) taxa abundance and (*b*) biovolume at the date of sampling during this study. (Online version in colour.)

feeders [17–19] and in some cases even carnivores [20]. Similarly, copepods and cladocerans can perform different feeding strategies including, among others, feeding-current and ambush feeding [21], thereby using a wide spectrum of resources. Consequently, trophic niche partitioning at a high phylogenetic level, such as class or family, will underestimate the trophic niche diversity in zooplankton guilds.

Traditionally, method limitations have made it difficult to assess trophic niche differences between zooplankton species. Experiments to estimate the grazing impact of zooplankton are time-consuming and elaborate, and the amount of biological material required for biogeochemical tracer studies on plankton communities [22] often exceeds what is feasible to sort out from diverse zooplankton samples. Consequently, few species are often included in these studies, limiting the ability to describe key niche differences between them. Similarly, the diet spectrum in plankton experimental studies is often limited to a small number of *a priori* hypothetical prey species and may not display the food web's full complexity with enough resolution. The challenges and limitations of studying feeding traits of zooplankton species have further created a biased knowledge towards larger organisms in the food web that are more frequently studied [23]. Since most of the zooplankton feeding studies are conducted at different sites and times of the year, often using different

methods, the comparison of existing information on zooplankton trophic niches is laborious. DNA metabarcoding of selected organisms has proven to be a useful tool for resolving trophic interactions [24,25] and is increasingly being used for studying trophic interactions of zooplankton [26–28]. With the high sensitivity of the polymerase chain reaction, metabarcoding requires very little biological material and is a non-*a priori* method with high taxonomic resolution. Metabarcoding allows for a food web-oriented approach as several zooplankton species can be investigated simultaneously [29], thereby providing detailed insights on trophic interactions and better linking the trophic niche diversity with energy flow.

Detailed knowledge about trophic interactions can be of particular importance for coastal ecosystems that experience a shift in phytoplankton community with an increase of cyanobacteria due to climate warming and eutrophication [30–32]. The Baltic Sea is a prime example, illustrating that nitrogen fixed by cyanobacteria supports the productivity of upper trophic levels [33,34]. Yet, as filamentous cyanobacteria are often considered unpalatable for copepods [35], the mechanism of trophic incorporation is not fully understood. While rotifers and microzooplankton are abundant in the Baltic Sea (figure 1), their potential trophic link with cyanobacteria is seldom considered. Without knowledge about

trophic partitioning between micro and mesozooplankton, the possible fate and sinks of cyanobacterial production in the plankton food web remain largely unknown.

In this study, we aimed to investigate the trophic niches of functionally diverse groups of zooplankton spanning both size and phylum. By sequencing *18S rRNA* and *16S rRNA* genes, we analysed zooplankton-associated prey of selected individuals of different size classes, including a ciliate, rotifers, copepods and cladocerans. The study was done at an offshore station in the Baltic Sea, where ciliates, rotifers and crustaceans at times dominate the zooplankton community. Our results show that the trophic niche diversity extends beyond broad phylogenetic groups and size classes and that small, rarely studied zooplankton fill an important role in the pathways of the coastal pelagic primary production.

## 2. Methods

### (a) Sampling

Zooplankton and water samples were collected at Landsort Deep monitoring station BY31 (58′35 N18′14 E) located in the eastern Baltic Sea proper, an offshore station at the deepest location of the Baltic Sea with 495 m depth. This frequently monitored offshore station experiences strong seasonal changes in biotic and abiotic properties (see electronic supplementary material, figures S1 and S2). To capture the seasonality of zooplankton (figure 1), samples were collected on 19 June and 15 August 2017, and on 16 March 2018, synchronized with the Swedish national pelagic monitoring programme [36].

Water samples, used for validation and to describe the potential zooplankton prey present in the water, were collected with 10 l Niskin bottles with 5 m depth intervals above the thermocline (0–30 m depth). The depths were mixed by adding an equal volume of water from the Niskin bottles; 1–3 l was sequentially filtered onto 25 mm diameter filters with 20 µm (nylon), 2 µm and 0.2 µm (polycarbonate) pore size. Filters were stored frozen at −80°C until further analysis. Zooplankton samples were collected with three vertical hauls from 0–30 m, 30–60 m and 60–100 m using a 90 µm WP2 closing plankton net (Hydrobios, Kiel, Germany). Ciliates were sampled with a 55 µm hand-towed plankton net in the upper 10 m layer (Hydrobios, Kiel, Germany). The zooplankton and ciliate samples were immediately preserved in 95% ethanol.

### (b) Zooplankton sorting and DNA metabarcoding

Individuals of abundant zooplankton species were identified under a stereomicroscope and selected from depth layers where they were most abundant (electronic supplementary material, table S1 and figure S2). This includes the rotifers *Synchaeta baltica*, *Synchaeta monopus* and *Keratella* spp., the cladocerans *Evadne nordmanni* and *Bosmina* spp., and the copepods *Temora longicornis*, *Acartia* spp., *Pseudocalanus* spp. and *Centropages hamatus*. All individual rotifers were rinsed five times in ethanol; crustaceans were rinsed five times in miliQ water and after that soaked for 30 s in a 1% bleach solution to remove contamination of external DNA. Five to 12 individuals from each species were randomly pooled into one sample tube and stored in 180 µl ALT lysis buffer (Qiagen, Hilden, Germany). A representative of the protozooplankton community, the ciliate *Helicostomella,* was transferred from the zooplankton samples onto a PET membrane-coated glass slide (Zeiss, Oberkochen, Germany) and covered with resin-based liquid cover glass (Zeiss). Single cells of *Helicostomella* were collected using a laser capture microdissection microscope (Zeiss) and 10–15 individuals per sample pooled into 10 µl ALT lysis buffer (Qiagen). All of the sorted zooplankton samples

were prepared in at least five replicates that were treated separately in all downstream analyses.

In the DNA metabarcoding analysis, we amplified a 500 bp long fragment of the V3–V4 region of the *16S rRNA* gene (*16S*) using universal primers 341F and 805R targeting both bacterial and plastidial *16S* of phototrophic eukaryotes [37,38], and a 400 bp long fragment of the V4 region of the *18S rRNA* gene (*18S*) using the primers 528F and 706R [39]. The amplicon libraries were sequenced on MiSeq (MSC 2.5.0.5/RTA 1.18.54) pair-end set-up (2 × 300 bp, v. 3, Illumina, San Diego, California). DNA sequences and associated metadata were uploaded to the European Nucleotide Archive (ENA) under accession no. PRJEB39191.

*16S* sequences were assigned to a custom-made database combining the SILVA *16S* reference database [40] with the PhytoREF database [41] to achieve an adequate taxonomic resolution for both prokaryotes and photoautotrophic eukaryotes. *18S* sequences were assigned to the Protist Ribosomal Reference database [42]. Details of sample processing, sequencing and bioinformatic analysis can be found in the electronic supplementary material.

### (c) Data analysis and visualization

Data filtering and statistical analysis were facilitated by the Phyloseq R package [43]. All sequences originating from the respective zooplankton consumer species in each sample were removed before data visualization. Heterogeneous sequencing depth was controlled for using subsampling (rarefaction) and subsequent conversion to relative abundance. Non-metric multi-dimensional scaling plots were based on Bray–Curtis distances and calculated with the 'metaMDS' function in the Vegan R package [44]. We used Bray–Curtis similarity index to assess diet overlap between samples (1-Bray–Curtis distance). Differences in the proportion of the specific diet of consumers were modelled with β regression using the 'betareg' function in R. We used the 'simper' function in the Vegan package to decide which prey species contributed most to the differences in diet overlap.

Figures were made using the ggplot2 R package [45]. The most important prevalent taxa (determined as taxa occupying at least 0.1 per cent of the sequences in at least 70 per cent of the samples in each sample group) were visualized in bipartite networks made in the Circlize R package [46]. All data used for the statistical analysis and plotting together with the R scripts to generate the figures were uploaded to the Dryad Digital Repository [47].

## 3. Results

### (a) Diversity of biotic associations

The Illumina sequencing effort produced over 37 million sequence reads that passed quality control. The *16S rRNA* gene (*16S*) that targets bacteria and photoautotrophic eukaryotes (plastids), generated 1492 unique ribosomal sequence variants (RSVs) of which 988 were found in the bulk water samples and 996 found in the selective zooplankton samples. The *18S rRNA* gene (*18S*) that targets all eukaryotes generated 3267 RSVs, of which 2258 were in the bulk water samples and 1394 found in the zooplankton samples. We found a broad range of organisms associated with the zooplankton organisms, including heterotrophic and autotrophic bacteria, phytoplankton, protozoans and metazoans.

We found that, on average, 85% of the *16S* sequence reads associated with the zooplankton samples were proteobacteria, which varied between zooplankton species and

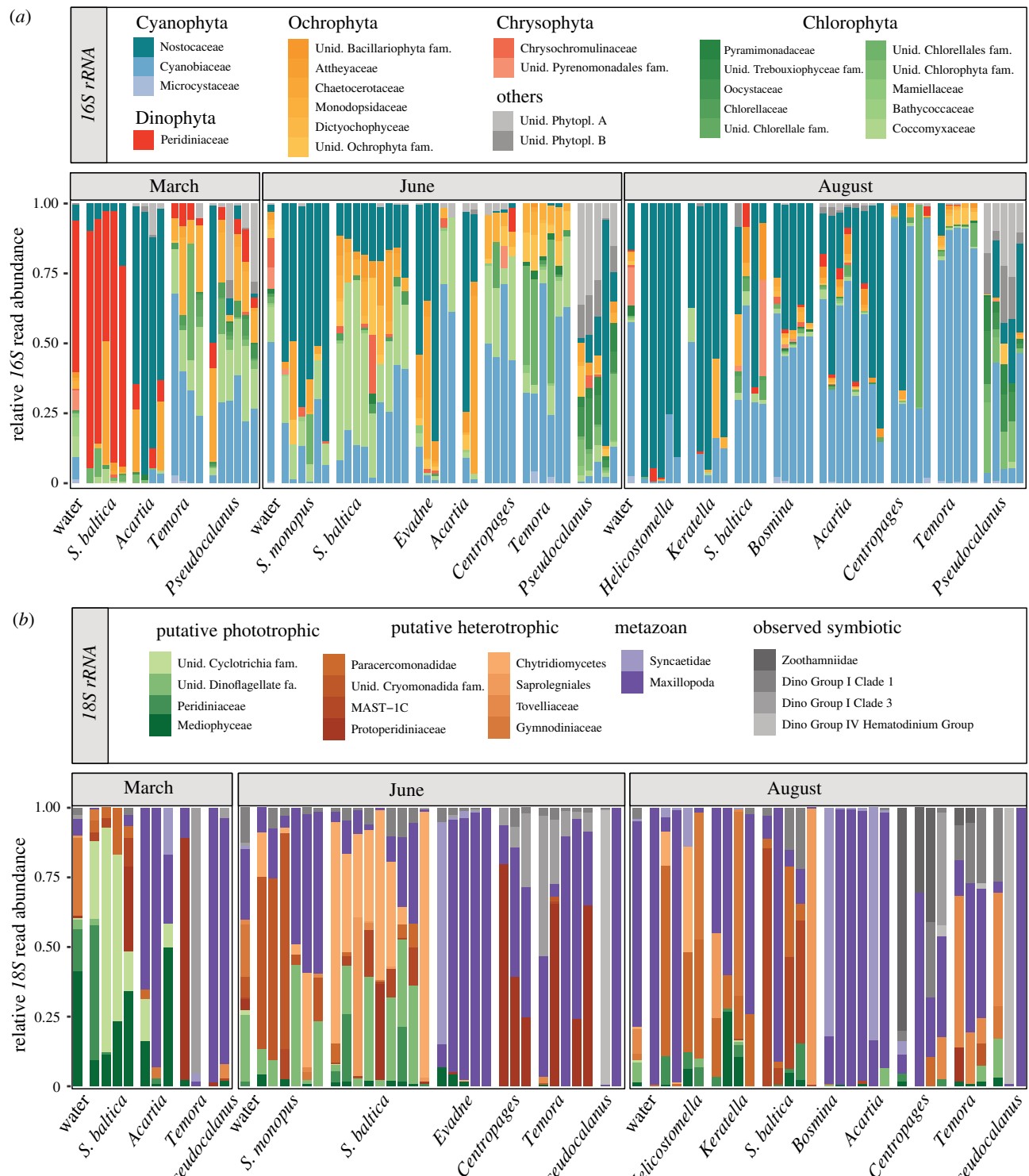

**Figure 2.** Relative abundance of sequence counts per family of (*a*) *16S rRNA* gene reads (photoautotrophic organisms only) and (*b*) *18S rRNA*, for different zooplankton consumer species and months in the Baltic Sea. The bars represent unique biological replicates. (Online version in colour.)

season (electronic supplementary material, appendix S1 and figure S4). Among photoautotrophic taxa (cyanobacteria and plastic-containing eukaryotes), associations of zooplankton consumer samples were dominated by cyanobacteria, green algae (Chlorophyta), diatoms (Bacillariophyta) and dinoflagellates (Dinophyceae) (figure 2*a*). Based on the *18S* reads, the zooplankton species were associated with a diversity of eukaryotic organisms, comprising both photoautotrophic and heterotrophic plankton and a diversity of potential symbiotic or parasitic organisms with oomycetes and dinoflagellates (figure 2*b*).

## (b) Trophic niche diversity in spring

During the spring months, from March to June, the rotifer *Synchaeta baltica* was the dominating zooplankton species in the Baltic Sea proper, accompanied by less abundant copepod species (figure 1*a*). The main primary producers were bloom-forming dinoflagellates and diatoms, but also the mixotrophic ciliate *Myrionecta* (figure 1*b*). In March, at the beginning of spring bloom, diet overlap between the zooplankton species was relatively low, according to the *16S* reads. The rotifer *S. baltica* had a diet overlap between 0.1 and 0.17 with the copepod groups, while the highest overlap

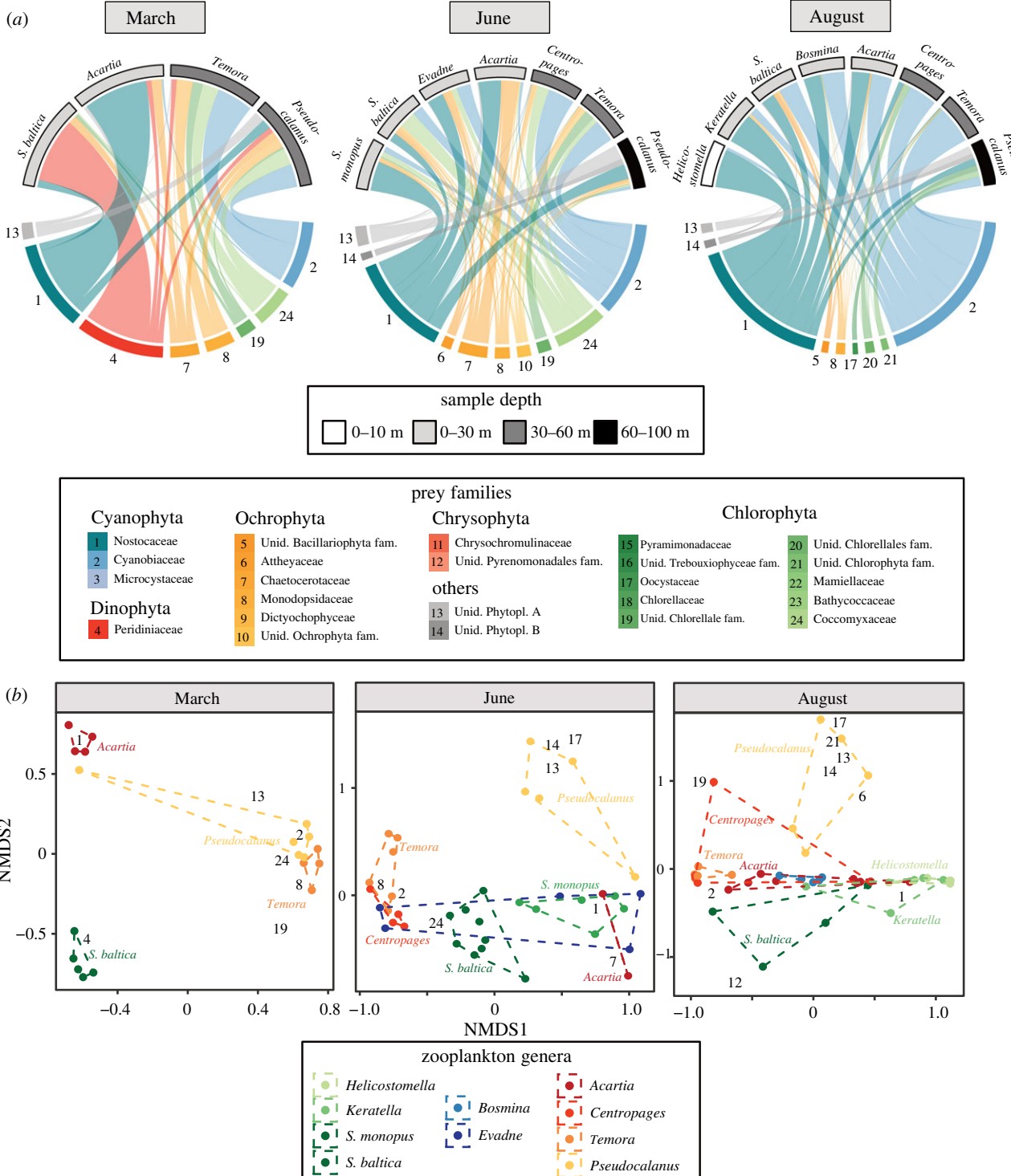

**Figure 3.** (a) Zooplankton consumer species (upper) with their most prevalent prey families (lower) based on *16S rRNA* gene reads. The thickness of the bars is proportional to relative rRNA read abundance. (b) Non-metric multidimensional scaling plot of Bray–Curtis distances between zooplankton samples (represented by coloured points) based on their prey (*16S rRNA* reads). The prey families responsible for the largest percentage of dissimilarity between any pair of zooplankton species are represented as numbers.

in diet was found between the copepods *Temora* and *Pseudocalanus* (0.53) (figure 3b; electronic supplementary material, figure S5). The rotifer *S. baltica* was mainly associated with the bloom-forming dinoflagellate *Peridiniella* (occupying on average 76% of the *16S* reads) (figure 3a). The copepods *Temora* and *Pseudocalanus* were associated with fewer sequences of *Peridiniella* compared to the rotifer (on average 6% of *16S* reads, d.f. = 3, $z = 16$, $p < 0.001$), but instead associated with various groups of small phytoplankton and picocyanobacteria. In March, *Acartia* was almost exclusively

associated with filamentous cyanobacteria (figure 3a). The *18S* sequences supported the association between *S. baltica* and *Peridiniella* but also revealed associations with the ciliate *Myrionecta*. The *18S* sequences further revealed associations between all zooplankton species and diatoms (figure 2b).

*Synchaeta baltica* reached its peak abundance in the Baltic Sea towards the end of the spring, in June, coordinated with the decline of dinoflagellates (Miozoa) (figure 1). Diet overlap between zooplankton species became more apparent but did not cluster according to phylogenetic affiliation. In June,

*S. baltica* had an equally high diet overlap with the copepod *Centropages* (0.48) and the cladoceran *Evadne* (0.39), compared to the sister species *S. monopus* (0.40) (figure 3*b*; electronic supplementary material, appendix S1 and figure S3). Similarly, the copepod *Acartia* had a higher diet overlap with *S. monopus* (0.53) than with the other copepods (overlap of 0.11 with *Temora*) (electronic supplementary material, figure S5). At the end of spring, cyanobacteria became more apparent in the diet of the rotifers, indicating a transition from a spring to a summer prey community (figure 3*a*).

## (c) Trophic niche diversity in summer

In August, the abundance and diversity of crustacean zooplankton increased in the Baltic Sea, and *Keratella* was the most abundant rotifer. The rotifer *Synchaeta baltica* was still present but with low abundance (figure 1*a*). The primary production was characterized by extensive blooms of filamentous cyanobacteria (figure 1*b*). In August, a large part of the variation in zooplankton diet read abundance could be explained by the abundance of filamentous cyanobacteria (Nostocaceae) and picocyanobacteria (Cyanobiaceae) (figure 3*a*).

The highest diet overlap was found between the heterotrophic ciliate *Helicostomella* and the rotifer *Keratella* in August (0.75), as they were mostly associated with filamentous cyanobacteria (occupying 93% and 74% of *16S* reads, respectively) (figure 3; electronic supplementary material, figure S5). *Synchaeta baltica* together with the cladoceran *Bosmina* and the copepod *Acartia* were associated with a lower proportion of filamentous cyanobacteria than the *Keratella* and *Helicostomella* (on average 39%, d.f. = 4, $z = 5.7$, $p < 0.001$), but with a larger proportion of picocyanobacteria (50%, d.f. = 5, $z = 5.7$, $p < 0.001$) as well as diverse small phytoplankton. Thus, the diet overlap between the two rotifer species *Keratella* and *S. baltica* was lower (0.42) than the overlap both between *Keratella* and the heterotrophic ciliate *Helicostomella* (0.75) and between *S. baltica* and the copepod *Acartia* (0.54). The copepods *Temora* and *Centropages* were associated with a low proportion of filamentous cyanobacteria (8%) and were almost exclusively associated with a higher relative proportion of picocyanobacteria (80%) compared with *Acartia*, *Bosmina* and *S. baltica* (d.f. = 5, $z = 4.6$, $p < 0.001$). Consequently, the copepod *Acartia* had a higher diet overlap with the cladoceran *Bosmina* (0.70) than with the other copepods (e.g. *Temora*, 0.52). Finally, *Pseudocalanus*, clustering alone, was associated with a significant proportion of unclassified organisms (up to 31% of *16S* reads) (figure 3*a*).

The *18S* sequences revealed various groups of heterotrophic flagellates associated with *S. baltica*, *Keratella* and *Helicostomella*. Small phytoplankton (chlorophytes and eustigmatophytes), heterotrophic protozoans of different phyla, as well as metazoans dominated the *18S* sequences of the cladocerans and copepods in summer (figure 2*b*).

# 4. Discussion

## (a) Niche diversity and overlap

In order to resolve the trophic niche diversity of zooplankton, we analysed the physical associations of several micro and mesozooplankton species using *18S* and *16S* rRNA gene sequencing of selected zooplankton. The trophic position and role of zooplankton in food webs are often derived from taxonomic groupings and size estimation [12,48]. However, our results highlight that clustering zooplankton by size or phylogeny does not capture the true differences in diet niche and leads to an underestimation of the trophic niche diversity of primary consumers in the pelagic food web. Rotifers in the Baltic Sea are often grouped with microzooplankton and are referred to as obligate filter feeders [49,50]. Despite this, we can not find support for a high diet overlap or clustering between rotifer species (figure 3*b*). Instead, our study indicates that *Synchaeta baltica* (approx. 350 µm) has a trophic niche more similar to cladocerans and copepods than to the other rotifers, *S. monopus*, *Keratella* and the ciliate *Helicostomella*. The diet of *S. baltica* in spring included bloom-forming phytoplankton taxa, including the dinoflagellate *Peridiniella* (Peridiniaceae, 20–35 µm) and the mixotrophic ciliate *Myrionecta* (Cyclotrichia, 45–55 µm) (figure 3), findings that are in line with previous studies that have observed predation on large phytoplankton and protozoa up to 50 µm by *Synchaeta* [16–19].

In contrast with *Synchaeta*, the smaller sized rotifer *Keratella* peaks in abundance during the summer (figure 1) and was mainly associated with larger filamentous cyanobacteria (Nostocaceae). *Keratella* revealed a higher diet overlap with the tintinnid ciliate *Helicostomella* than with *S. baltica* (figure 3). The size of the *Keratella* (150 µm) and *Helicostomella* (100 µm) compared to cyanobacteria filaments that often exceed 1 mm suggests that these consumers do not feed directly on filamentous cyanobacteria, but rather on degraded filaments. This is supported by experiments indicating a filter-feeding behaviour of *Keratella* [15] that prefers partially degraded food (detritus) over living cells [51]. The same can be expected for *Helicostomella*, although few studies have investigated the selectivity of this ciliate [52–54]. The detritivorous feeding niche of *Keratella* and *Helicostomella* suggested here is further supported by a relatively high proportion of associated crustacean DNA (figure 2*b*), which for similar reasons is unlikely to be preyed upon directly and is likely ingested in the form of particulate organic matter. We suggest that filamentous cyanobacteria likely contribute to a pool of organic matter that is both available and attractive for detritivorous rotifers and ciliates. The results point to the importance of grouping zooplankton according to function rather than taxa, with some rotifers occupying the function of selective feeders similar to copepods, a distinction already proposed in a study by Arndt in 1993 [15].

Similar to rotifers, there was no clear clustering within copepod and cladoceran species based on their prey composition. While *Temora* and *Centropages* shared high diet overlap, mostly associated with picocyanobacteria, *Acartia* had a diet more similar to the cladocerans and the rotifer *Synchaeta baltica* during the summer months, relying on a large diversity of resources, including both filamentous and picocyanobacteria (figure 3). On the other side, the copepod *Pseudocalanus* occupies its own niche, feeding in all seasons on various unclassified organisms. The different feeding niches of copepods may reflect their vertical distribution (electronic supplementary material, figure S2), as *Temora* and *Centropages* are more abundant at 30–60 m depth compared to *Acartia* that dominates in the upper 30 m. By contrast, *Pseudocalanus* extends deeper in the water column compared to the other zooplankton [55,56]. The association of *Pseudocalanus* with unidentified taxa might indicate that the components of its natural diet are not well represented

*Proc. R. Soc. B* **288**: 20210908

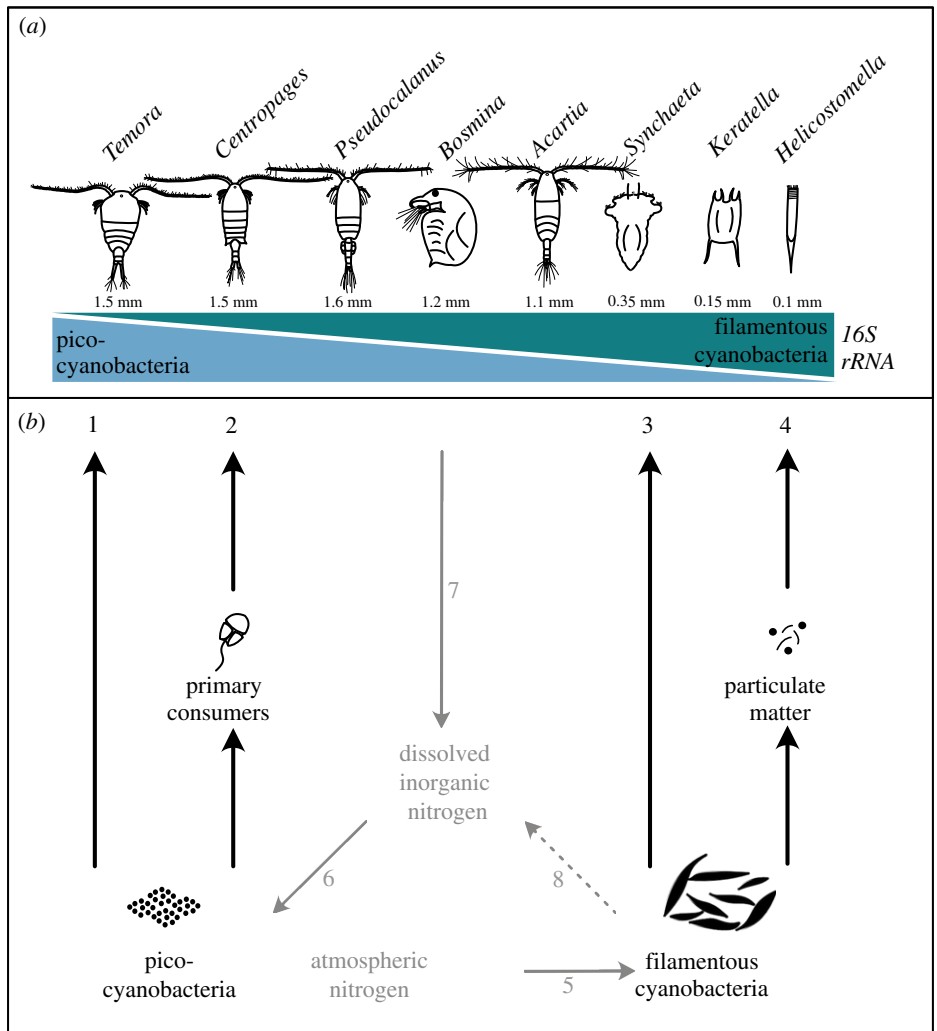

**Figure 4.** Illustration of dominant zooplankton consumers and alternative energy transfer pathways in the pelagic food web. (*a*) Size range of selected zooplankton species aligned with the relative read abundance of associated pico- and filamentous cyanobacteria. (*b*) Black arrows illustrate alternative pathways of energy transfer from primary producers to zooplankton consumers. Ingestion of picocyanobacteria can be directly (1) or via primary consumers (2). Filamentous cyanobacteria can be ingested alive (3) or in the state of decay (4). Shaded arrows denote putative pathways of nitrogen fixed by filamentous cyanobacteria (5) that may enter the food web either via consumption (3, 4) but may also stimulate the production of picocyanobacteria (6). Grazers of filamentous cyanobacteria may enhance the release of dissolved inorganic nitrogen (7), in contrast with previously suggested passive leaking (8). (Online version in colour.)

in the taxonomic databases. The differences in trophic niches of copepods are supported by a more intense seasonal sample analysis of crustacean zooplankton described in the previous study [29] and more extensive sampling several locations in the Baltic Sea proper (Baptiste Serandour 2021, pers. comm).

## (b) Food web implications

The trophic niche partitioning of zooplankton has implications in the food web dynamics in spring and summer. As the decline of dinoflagellates in spring coincides with the peak of *Synchaeta* (figure 1), grazing by rotifers should be considered a potential cause of the phytoplankton spring bloom decline in addition to nutrient limitation in the upper water column [57]. This is further supported by a study from the Mediterranean Sea, where *Synchaeta* was estimated to consume up to 80% of the daily production of a dinoflagellate bloom [58]. Given that cladocerans and copepods are temporally decoupled from the spring bloom (figure 1), *Synchaeta* is likely a major pathway of energy

transfer to higher trophic levels in the Baltic Sea and possibly in other ecosystems where this species is abundant [58].

Pico and filamentous cyanobacteria are favoured under climate warming and eutrophication and are increasing in both marine and freshwater systems [30,59]. Due to their significance as food resources in the summer community of the Baltic Sea [33,49,60], understanding the pathways of cyanobacteria incorporation into food webs is important. The trophic role of filamentous cyanobacteria is widely debated as they are the main source of nitrogen fixation in the Baltic Sea, which is suggested to be nitrogen limited in summer [61]. Several studies suggest little or no grazing on filamentous cyanobacteria by zooplankton [34,62,63], but our study shows that the microzooplankton *Helicostomella* and *Keratella*, but also to some extent *Bosmina* and *Acartia* grazed on filamentous cyanobacteria, either living or degraded.

Zooplankton feeding on filamentous cyanobacteria may also act as important vectors of diazotroph nitrogen availability to upper trophic levels by stimulating the microbial food web. This is supported by an experiment by Arndt [15] showing that the presence of filter-feeding *Keratella*

stimulates the growth of both heterotrophic flagellates and bacteria. Arndt proposed that *Keratella*, through its feeding, enhances leaking of dissolved matter from the algae [4], thereby supporting the increased biomass of both bacteria and protozoa. Tracer studies suggest that diazotroph nitrogen is mainly incorporated in the Baltic Sea food web by passive leaking of nitrogen compounds by filamentous cyanobacteria [64,65] that stimulate the production of heterotrophic bacteria and picocyanobacteria [33,34,66]. Our study confirms that picocyanobacteria are a key resource for the larger zooplankton species, primarily *Temora* and *Centropages*, but also, to some extent, *Acartia*, *Bosmina* and *Synchaeta* (figure 4*a*), and are as such indirectly supported by filamentous cyanobacteria. In addition to passive leaking, feeding by copepods, cladocerans, rotifers and ciliates on filamentous cyanobacteria actively enhances leaking of diazotroph nitrogen. This constitutes an alternative pathway of cyanobacteria incorporation into the pelagic food web and enhances the support for copepods that rely on the microbial food web (figure 4*b*).

While DNA metabarcoding has become more frequently used over the last decade, few studies have until now exploited the potential of investigating the feeding niche diversity of the entire zooplankton community spanning several phyla and size classes. We show that the method used here is applicable for several metazoan taxa and potentially protozooplankton, using the ciliate *Helicostomella* as an example. DNA metabarcoding could be relevant in future investigations to unveil the role of rarer species and better comprehend the ecosystem function. Our approach shows the advantage of investigating the prey composition of diverse species in natural systems with DNA metabarcoding that reveals the entire food spectrum. By putting weight on the relative comparison between zooplankton species, we could capture key differences in zooplankton resource. While we can discuss possible ecosystem effects of diverse zooplankton feeding, the metabarcoding data is inevitably proportional. Thus, the data do not reveal information about feeding rates or biomasses. Despite this, metabarcoding has the potential to serve as an important complement to food web models that implement population biomasses and metabolic energy demands [67], by bringing details of species-specific feeding interactions to the model.

Our results highlight a large variation in resource use between groups of zooplankton that may stabilize energy transfer in food webs by pathways of energy flow that are rarely described, particularly during seasons when primary producers include pico and filamentous cyanobacteria. The presence of multitrophic species with the ability to prey on different food web components may contribute to ecosystem resilience. We emphasize the importance of understanding the trophic niche diversity of key zooplankton taxa to generate an accurate understanding of ecosystem functioning. Food web models based on size or phylogeny may not capture the important role of individual species and may not be detailed enough to predict energy pathways of plankton food webs and thus the vulnerability of ecosystems to environmental change. Combined with estimates of prey biomass and predator feeding rates, the approach used in this study is a suitable entry point to food web modelling and ecosystem network analysis.

**Data accessibility.** DNA sequences and associated metadata were uploaded to the ENA under accession no. PRJEB39191 (https://www.ebi.ac.uk/ena/browser/view/PRJEB39191). RSV data files used for statistical analysis, together with the R code to generate the figures are available from the Dryad Digital Repository: https://doi.org/10.5061/dryad.gb5mkkwpw [47].

**Authors' contributions.** A.N.: conceptualization, data curation, formal analysis, investigation, methodology, project administration, software, validation, visualization, writing-original draft, writing-review and editing; S.Z.-T: conceptualization, data curation, investigation, methodology, project administration, validation, writing-review and editing; M.W.: conceptualization, funding acquisition, project administration, resources, supervision, writing-review and editing.

All authors gave final approval for publication and agreed to be held accountable for the work performed therein.

**Competing interests.** We have no competing interests.

**Funding.** The research was funded by the Swedish Research Council project no. 2016-04685.

**Acknowledgements.** We would like to direct our gratitude to the pelagic monitoring group at Stockholm University, especially Stefan Svensson and Jakob Walve for hosting us during the sampling effort. We also acknowledge support from the National Genomics infrastructure in Stockholm and Uppsala, funded by Science for Life Laboratory, the Knut and Alice Wallenberg Foundation and the Swedish Research Council, and SNIC/Uppsala Multidisciplinary Centre for Advanced Computational Science for assistance with massively parallel sequencing and access to the UPPMAX computational infrastructure.

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
