## [Peer Review File · Proceedings of the Royal Society B: Biological Sciences]

Review History

RSPB-2021-0348.R0 (Original submission)

Review form: Reviewer 1

Recommendation

Accept with minor revision (please list in comments)

Scientific importance: Is the manuscript an original and important contribution to its field?
Excellent

General interest: Is the paper of sufficient general interest?
Good

Quality of the paper: Is the overall quality of the paper suitable?
Good

Is the length of the paper justified?
Yes

Should the paper be seen by a specialist statistical reviewer?
No

Do you have any concerns about statistical analyses in this paper? If so, please specify them explicitly in your report.

No

It is a condition of publication that authors make their supporting data, code and materials available - either as supplementary material or hosted in an external repository. Please rate, if applicable, the supporting data on the following criteria.

Is it accessible?

Yes

Is it clear?

N/A

Is it adequate?

N/A

Do you have any ethical concerns with this paper?

No

Comments to the Author

The authors developed a metabarcoding analysis to explore the seasonal trophic dynamics among plankton organisms in a coastal environment of the Baltic Sea. The study is time-needed and provides important insights on species trophic niches, information that be applied for other similar ecosystems as well. The title of the study is informative and suitable, and the authors provide a well-written summary of their work in the abstract. The manuscript is coherent with comprehensive figures. Overall, I propose minor corrections which in my opinion will improve the manuscripts' readability and highlight the significance of this study. Please find below my comments and suggested corrections.

Major suggestions:

1. Lines 47-48: Please rephrase this sentence. Species are multifunctional and therefore can fit in different groups depending on their traits (e.g., in lines 48-53 the authors provide examples of how different taxonomic species can be grouped together based on their feeding traits). As authors mention in lines 65-66, is the limited quantified and qualified information that causes the misrepresentation of the diverse trophic niche and not the approaches per se.
2. To improve readability, move lines 55-64 after line 83 and link the sentences. For example:
 "[...] can be investigated simultaneously [35]. Metabarcoding can therefore provide significant insights on trophic interactions and better link the trophic niche diversity with energy flow. This can be of particular importance for coastal ecosystems with high ecosystem services that experience an increase in pico and filamentous cyanobacteria due to climate warming and eutrophication [21-23]. [...]"
3. Move lines 83-85 to the discussion and expand with one sentence saying what is the main reason/limitation why there are only a few studies.
4. Lines 87-90: This sentence is not well fitted with the intro. I propose to change added as a result output after line 93.
 "[...] Our results show that the trophic niche diversity extends beyond broad phylogenetic groups and size classes, and that rarely studied microzooplankton (rotifers and ciliates) fill an important role in the pathways of the coastal pelagic primary production."
5. Sample region: Please add one sentence about why this location is chosen and the ecosystem properties of the sampling region (e.g., nutrient). Also please include a map in the Appendix.
6. Lines 270-272: This is a very interesting statement. Please provide more information with the relevant key message from the study [35] and how salinity is a potential driver.
7. Lines 289-292: This sentence is confusing as it includes copepods and protozoans. If you

want to make this sentence copepods specific please remove the “our study shows that primarily smaller zooplankton, such as *Helicostomella* and *Keratella*” otherwise rephrase the whole sentence.

8. Lines 315-318, 325-329: Please move those statements in the discussion and expand further. In lines 325-329, the authors made an excellent point of how our modelling projections on energy flow would benefit by including trophic niches. But, as it has been mentioned in lines 315-318, metabarcoding cannot offer qualified data useful for models and data (e.g., stoichiometry, biomass, rates) from other approaches need to be combined. It would be helpful for the community if the authors can be more specific on the main limitations of their approach and which kind of data are necessary for implementing their data into models and which model design they think that are most suitable.

Minor:

1. lines 31-33: Please rephrase the first part of the sentence as is identical to the first sentence of the abstract.
2. lines 45-47: This sentence needs to be rephrased. “True” is a strong word in the “true ecosystem structure”, please remove it. Additionally, like every theory, the trait-theory has its strengths and limitations, therefore the research question and model design will define if a trait-based approach is the best option.
3. line 92: change to “a coastal region of the Baltic Sea”.
4. replace articles with studies throughout the discussion.
5. Please include the DOI for all references and write in italics the name of species.

Review form: Reviewer 2

Recommendation

Accept with minor revision (please list in comments)

Scientific importance: Is the manuscript an original and important contribution to its field?

Good

General interest: Is the paper of sufficient general interest?

Good

Quality of the paper: Is the overall quality of the paper suitable?

Good

Is the length of the paper justified?

Yes

Should the paper be seen by a specialist statistical reviewer?

No

Do you have any concerns about statistical analyses in this paper? If so, please specify them explicitly in your report.

Yes

It is a condition of publication that authors make their supporting data, code and materials available - either as supplementary material or hosted in an external repository. Please rate, if applicable, the supporting data on the following criteria.

Is it accessible?

Yes

Is it clear?

Yes

Is it adequate?

Yes

Do you have any ethical concerns with this paper?

No

Comments to the Author

The manuscript works on the diet preference to micro and mesozooplankton in the Baltic. The paper is clear and well written, with a few exceptions (indicated below).

There are some changes or additional information that, if provided, would improve the quality of the manuscript. I have also some statistical concerns on the ANOVA, since the authors are analyzing the same data twice. Also about the adequacy of ANOVA for this analyses.

In Methods: Why the ciliates were sampled only from the top 10m? Wouldn't this make the study don't match each part? Could this impact how the results from Ciliates are interpreted, since they are only pulled from the surface.

Line 143: BCI is then Bray-Curtis similarity? I suppose metaMDS calculated BC dissimilarity index (that I think that was the original definition, as dissimilarity, ranging from 0 for identical, to 1 with all non overlapping) and then the authors transformed it into the BC similarity index (1-dissimilarity) and multiplied by 100? I am truly asking, since I have not seen this BCI before, and I think is just BC similarity (that the authors might refer then better as: "We used Bray-Curtis similarity index (in percentage) as an estimate to assess diet overlap between samples". I am not sure why the multiplying by 100, instead of leaving is as is. I see this part potentially confusing for the reader without expertise in this kind of indices.

Line 144: "Differences in diet overlap and differences in the proportion of the specific diet of consumers" but these two are the same: diet overlap is Bray-Curtis similarity index derived from the proportion of taxa. I see how the authors used them, but just to be aware that they are the same analyses based in the same data. These analyses are carried out with an ANOVA by ranks, that I do not think it solves the problem of the fact that the data the authors are analyzing are percentages. Maybe a PERMANOVA would be more adequate for the BC index, and chi-square with the relative abundances?

There are some other things that would make this paper more solid:

- 1) Could the authors provide a vertical profile plots of the Chl-a, temp and salinity of the water column? This would help understanding seasonality changes.
- 2) Is there a possibility that the different rotifer species show vertical partition, with then the associated changes in diet (that is suggested for copepods)? Again, without an explanation of the water column might be tricky. And since all niskin samples are mixed later on, it might be difficult to interpret, but it should be at least discussed.
- 3) Did the authors mixed also the two depths of the zooplankton? Are there differences in community between both regarding both the rotifers and the copepods? Any information from this would also help discussing the differences between rotifers and copepods.

Minor:

Line 140: maybe better using "heterogeneous" instead of "heterogenic" (this is a term for allele

studies).

Review form: Reviewer 3

Recommendation

Accept with minor revision (please list in comments)

Scientific importance: Is the manuscript an original and important contribution to its field?

Excellent

General interest: Is the paper of sufficient general interest?

Good

Quality of the paper: Is the overall quality of the paper suitable?

Excellent

Is the length of the paper justified?

Yes

Should the paper be seen by a specialist statistical reviewer?

No

Do you have any concerns about statistical analyses in this paper? If so, please specify them explicitly in your report.

No

It is a condition of publication that authors make their supporting data, code and materials available - either as supplementary material or hosted in an external repository. Please rate, if applicable, the supporting data on the following criteria.

Is it accessible?

Yes

Is it clear?

Yes

Is it adequate?

Yes

Do you have any ethical concerns with this paper?

No

Comments to the Author

Phylogeny and body size are the traditional theories to investigate trophic interactions and predict food web dynamics and ecosystem functioning. In this work, Novotny, Zamora-Terol and Winder highlight the benefit of using DNA metabarcoding to investigate trophic interaction in a pelagic food web in the Baltic Sea, focusing on the functional diversity. Indeed DNA metabarcoding can describe the functional diversity in the food web. With this method, the authors were able to identify alternative pathways of energy, which stabilize the pelagic food web. As such, the topic and discussion brought by this article is very valuable. The paper is well written, and I found minor points to be unclear:

- The authors bring a valuable point towards the interesting functions of rotifers and ciliates in trophic niche diversity, which conflicts with the main theories of trophic interaction based on phylum and size (l. 232-235). This conclusion is based on “selected zooplankton” with one species of ciliates and three species of rotifers, which -I believe- was dependent on the abundance of these organisms. The manuscript would benefit with additional information about how this method could be relevant for rarer species.
- In the introduction (l. 65-69), the authors argued that metabarcoding would be more efficient than experiments (e.g. dilution techniques). However, the authors concluded that this method “combined with validating experiments” would be suitable method for food web modeling and ecosystem networking analysis. The conclusion would be more informative with an example of such validating experiment.
- In the results section. According to the statistical guidelines, the authors should add the degree of freedom of the Kruskal-Wallis test; and use the lowercase p for the p.value.
- L.101. I understood that water samples served as control; it is better to state it clearly.
- L.103. The water was filtered “sequentially” from the smaller to larger pore size. Should not be the reverse order?
- L.160-161. I do not understand how the number of RSVs in the bulk water samples (2771) and in the selective zooplankton samples (1799) can be larger than the total (1483).
- L.166. I believe it is better to be consistent between the text and the figures. Here, in the text, the authors refer to 90% of heterotrophic bacteria; while in Figure S2 they refer to 85% of Proteobacteria.
- L.176. I was confused with the sudden division into spring vs summer because there are three dates of observations. Then, I understood that the results were referring to the first observation (i.e., march). Consider replacing “In spring” by “In March”.
- L.182. Figure 3a instead of figure 4?
- L.192. The decline of Peridiniella is not shown in Figure 1. Please, mention the phylum corresponding to Peridiniella.
- L.206. For consistency, please refer to cyanobiaceae (?)
- L.260. There is a problem with the reference Arndt (1993)
- L.312-314. Few elements reduced the clarity of the key message. Please, consider removing “rather than focusing on the sequence read abundance of prey” because the relative comparison is using the read abundance; and “without the assumption of a relationship between read-count and biomass” as the authors state that they do not have such information (l.317).
- Figure 1a. There are 8 dynamics shown and 7 with the legends.
- Figure 3. To link the panels a and b, I suggest to have the same numbered labels.
- Figure 4. Information about the size of the organisms is only mention in the discussion. As the authors aimed at showing differences between phylum/size and metabarcoding method, the authors could mention the size in this (nice!) figure 4a summarizing the results/discussion.

Decision letter (RSPB-2021-0348.R0)

19-Mar-2021

Dear Mr Novotny:

I am writing to inform you that your manuscript RSPB-2021-0348 entitled "DNA metabarcoding reveals trophic niche diversity of micro and mesozooplankton species" has, in its current form, been rejected for publication in Proceedings B.

This action has been taken on the advice of referees, who are positive overall but have recommended that substantial revisions are nevertheless necessary. With this in mind we would be happy to consider a resubmission, provided the comments of the referees are fully addressed. However please note that this is not a provisional acceptance.

Sincerely,
 Professor Hans Heesterbeek
 mailto: proceedingsb@royalsociety.org

Associate Editor
 Board Member: 1
 Comments to Author:

We have now obtained three expert reviews of your manuscript, and I am glad to tell you that all the reviewers were positive about your study and I agree that this is an important contribution to better understand complex food web pathways. Two of the reviewers ask for clarifications on different aspects as well as suggest improvements on the flow and clarity of the paper. Reviewer 2 though has questions about your statistical analysis as well as sampling method, among others, which would be important to address.

Reviewer(s)' Comments to Author:

Referee: 1

Comments to the Author(s)

The authors developed a metabarcoding analysis to explore the seasonal trophic dynamics among plankton organisms in a coastal environment of the Baltic Sea. The study is time-needed and provides important insights on species trophic niches, information that be applied for other similar ecosystems as well. The title of the study is informative and suitable, and the authors provide a well-written summary of their work in the abstract. The manuscript is coherent with comprehensive figures. Overall, I propose minor corrections which in my opinion will improve the manuscripts' readability and highlight the significance of this study. Please find below my comments and suggested corrections.

Major suggestions:

1. Lines 47-48: Please rephrase this sentence. Species are multifunctional and therefore can fit in different groups depending on their traits (e.g., in lines 48-53 the authors provide examples of how different taxonomic species can be grouped together based on their feeding traits). As authors mention in lines 65-66, is the limited quantified and qualified information that causes the misrepresentation of the diverse trophic niche and not the approaches per se.

2. To improve readability, move lines 55-64 after line 83 and link the sentences. For example: "[...] can be investigated simultaneously [35]. Metabarcoding can therefore provide significant insights on trophic interactions and better link the trophic niche diversity with energy flow. This can be of particular importance for coastal ecosystems with high ecosystem services that experience an increase in pico and filamentous cyanobacteria due to climate warming and eutrophication [21-23]. [...]"
3. Move lines 83-85 to the discussion and expand with one sentence saying what is the main reason/limitation why there are only a few studies.
4. Lines 87-90: This sentence is not well fitted with the intro. I propose to change added as a result output after line 93.
"[...] Our results show that the trophic niche diversity extends beyond broad phylogenetic groups and size classes, and that rarely studied microzooplankton (rotifers and ciliates) fill an important role in the pathways of the coastal pelagic primary production."
5. Sample region: Please add one sentence about why this location is chosen and the ecosystem properties of the sampling region (e.g., nutrient). Also please include a map in the Appendix.
6. Lines 270-272: This is a very interesting statement. Please provide more information with the relevant key message from the study [35] and how salinity is a potential driver.
7. Lines 289-292: This sentence is confusing as it includes copepods and protozoans. If you want to make this sentence copepods specific please remove the "our study shows that primarily smaller zooplankton, such as *Helicostomella* and *Keratella*" otherwise rephrase the whole sentence.
8. Lines 315-318, 325-329: Please move those statements in the discussion and expand further. In lines 325-329, the authors made an excellent point of how our modelling projections on energy flow would benefit by including trophic niches. But, as it has been mentioned in lines 315-318, metabarcoding cannot offer qualified data useful for models and data (e.g., stoichiometry, biomass, rates) from other approaches need to be combined. It would be helpful for the community if the authors can be more specific on the main limitations of their approach and which kind of data are necessary for implementing their data into models and which model design they think that are most suitable.

Minor:

1. lines 31-33: Please rephrase the first part of the sentence as is identical to the first sentence of the abstract.
2. lines 45-47: This sentence needs to be rephrased. "True" is a strong word in the "true ecosystem structure", please remove it. Additionally, like every theory, the trait-theory has its strengths and limitations, therefore the research question and model design will define if a trait-based approach is the best option.
3. line 92: change to "a coastal region of the Baltic Sea".
4. replace articles with studies throughout the discussion.
5. Please include the DOI for all references and write in italics the name of species.

Referee: 2

Comments to the Author(s)

The manuscript works on the diet preference to micro and mesozooplankton in the Baltic. The paper is clear and well written, with a few exceptions (indicated below).

There are some changes or additional information that, if provided, would improve the quality of the manuscript. I have also some statistical concerns on the ANOVA, since the authors are analyzing the same data twice. Also about the adequacy of ANOVA for this analyses.

In Methods: Why the ciliates were sampled only from the top 10m? Wouldn't this make the study don't match each part? Could this impact how the results from Ciliates are interpreted, since they are only pulled from the surface.

Line 143: BCI is then Bray-Curtis similarity? I suppose metaMDS calculated BC dissimilarity index (that I think that was the original definition, as dissimilarity, ranging from 0 for identical, to 1 with all non overlapping) and then the authors transformed it into the BC similarity index (1-dissimilarity) and multiplied by 100? I am truly asking, since I have not seen this BCI before, and I think is just BC similarity (that the authors might refer then better as: "We used Bray-Curtis similarity index (in percentage) as an estimate to assess diet overlap between samples". I am not sure why the multiplying by 100, instead of leaving is as is. I see this part potentially confusing for the reader without expertise in this kind of indices.

Line 144: "Differences in diet overlap and differences in the proportion of the specific diet of consumers" but these two are the same: diet overlap is Bray-Curtis similarity index derived from the proportion of taxa. I see how the authors used them, but just to be aware that they are the same analyses based in the same data. These analyses are carried out with an ANOVA by ranks, that I do not think it solves the problem of the fact that the data the authors are analyzing are percentages. Maybe a PERMANOVA would be more adequate for the BC index, and chi-square with the relative abundances?

There are some other things that would make this paper more solid:

- 1) Could the authors provide a vertical profile plots of the Chl-a, temp and salinity of the water column? This would help understanding seasonality changes.
- 2) Is there a possibility that the different rotifer species show vertical partition, with then the associated changes in diet (that is suggested for copepods)? Again, without an explanation of the water column might be tricky. And since all niskin samples are mixed later on, it might be difficult to interpret, but it should be at least discussed.
- 3) Did the authors mixed also the two depths of the zooplankton? Are there differences in community between both regarding both the rotifers and the copepods? Any information from this would also help discussing the differences between rotifers and copepods.

Minor:

Line 140: maybe better using "heterogeneous" instead of "heterogenic" (this is a term for allele studies).

Referee: 3

Comments to the Author(s)

Phylogeny and body size are the traditional theories to investigate trophic interactions and predict food web dynamics and ecosystem functioning. In this work, Novotny, Zamora-Terol and Winder highlight the benefit of using DNA metabarcoding to investigate trophic interaction in a pelagic food web in the Baltic Sea, focusing on the functional diversity. Indeed DNA metabarcoding can describe the functional diversity in the food web. With this method, the authors were able to identify alternative pathways of energy, which stabilize the pelagic food web. As such, the topic and discussion brought by this article is very valuable. The paper is well written, and I found minor points to be unclear:

- The authors bring a valuable point towards the interesting functions of rotifers and ciliates in trophic niche diversity, which conflicts with the main theories of trophic interaction based on phylum and size (l. 232-235). This conclusion is based on "selected zooplankton" with one species of ciliates and three species of rotifers, which -I believe- was dependent on the abundance of these organisms. The manuscript would benefit with additional information about how this method could be relevant for rarer species.
- In the introduction (l. 65-69), the authors argued that metabarcoding would be more efficient than experiments (e.g. dilution techniques). However, the authors concluded that this method "combined with validating experiments" would be suitable method for food web modeling and

ecosystem networking analysis. The conclusion would be more informative with an example of such validating experiment.

- In the results section. According to the statistical guidelines, the authors should add the degree of freedom of the Kruskal-Wallis test; and use the lowercase p for the p.value.
- L.101. I understood that water samples served as control; it is better to state it clearly.
- L.103. The water was filtered “sequentially” from the smaller to larger pore size. Should not be the reverse order?
- L.160-161. I do not understand how the number of RSVs in the bulk water samples (2771) and in the selective zooplankton samples (1799) can be larger than the total (1483).
- L.166. I believe it is better to be consistent between the text and the figures. Here, in the text, the authors refer to 90% of heterotrophic bacteria; while in Figure S2 they refer to 85% of Proteobacteria.
- L.176. I was confused with the sudden division into spring vs summer because there are three dates of observations. Then, I understood that the results were referring to the first observation (i.e., march). Consider replacing “In spring” by “In March”.
- L.182. Figure 3a instead of figure 4?
- L.192. The decline of Peridiniella is not shown in Figure 1. Please, mention the phylum corresponding to Peridiniella.
- L.206. For consistency, please refer to cyanobiaceae (?)
- L.260. There is a problem with the reference Arndt (1993)
- L.312-314. Few elements reduced the clarity of the key message. Please, consider removing “rather than focusing on the sequence read abundance of prey” because the relative comparison is using the read abundance; and “without the assumption of a relationship between read-count and biomass” as the authors state that they do not have such information (l.317).
- Figure 1a. There are 8 dynamics shown and 7 with the legends.
- Figure 3. To link the panels a and b, I suggest to have the same numbered labels.
- Figure 4. Information about the size of the organisms is only mention in the discussion. As the authors aimed at showing differences between phylum/size and metabarcoding method, the authors could mention the size in this (nice!) figure 4a summarizing the results/discussion.

Author's Response to Decision Letter for (RSPB-2021-0348.R0)

See Appendix A.

RSPB-2021-0908.R0

Review form: Reviewer 1 (Maria Grigoratou)

Recommendation

Accept as is

Scientific importance: Is the manuscript an original and important contribution to its field?

Excellent

General interest: Is the paper of sufficient general interest?

Good

Quality of the paper: Is the overall quality of the paper suitable?

Good

Is the length of the paper justified?

Yes

Should the paper be seen by a specialist statistical reviewer?

No

Do you have any concerns about statistical analyses in this paper? If so, please specify them explicitly in your report.

No

It is a condition of publication that authors make their supporting data, code and materials available - either as supplementary material or hosted in an external repository. Please rate, if applicable, the supporting data on the following criteria.

Is it accessible?

Yes

Is it clear?

Yes

Is it adequate?

Yes

Do you have any ethical concerns with this paper?

No

Comments to the Author

Dear authors,

I am happy with the revised version of the manuscript, thank you for implementing the suggested corrections. I reviewed your manuscript with great pleasure and I would like to thank you for bringing new insights on plankton dynamics with your study.

Review form: Reviewer 2

Recommendation

Accept as is

Scientific importance: Is the manuscript an original and important contribution to its field?

Good

General interest: Is the paper of sufficient general interest?

Good

Quality of the paper: Is the overall quality of the paper suitable?

Good

Is the length of the paper justified?

Yes

Should the paper be seen by a specialist statistical reviewer?

No

Do you have any concerns about statistical analyses in this paper? If so, please specify them explicitly in your report.

No

It is a condition of publication that authors make their supporting data, code and materials available - either as supplementary material or hosted in an external repository. Please rate, if applicable, the supporting data on the following criteria.

Is it accessible?

Yes

Is it clear?

Yes

Is it adequate?

Yes

Do you have any ethical concerns with this paper?

No

Comments to the Author

The authors have addressed all my concerns

Review form: Reviewer 3

Recommendation

Accept with minor revision (please list in comments)

Scientific importance: Is the manuscript an original and important contribution to its field?

Excellent

General interest: Is the paper of sufficient general interest?

Excellent

Quality of the paper: Is the overall quality of the paper suitable?

Excellent

Is the length of the paper justified?

Yes

Should the paper be seen by a specialist statistical reviewer?

No

Do you have any concerns about statistical analyses in this paper? If so, please specify them explicitly in your report.

No

It is a condition of publication that authors make their supporting data, code and materials available - either as supplementary material or hosted in an external repository. Please rate, if applicable, the supporting data on the following criteria.

Is it accessible?

N/A

Is it clear?

N/A

Is it adequate?

N/A

Do you have any ethical concerns with this paper?

No

Comments to the Author

First, I reviewed the responses provided regarding the comments. The authors replied thoroughly to all concerns, and included them in the manuscript.

Then, I reviewed the new version of the manuscript (i.e. without the track changes). I found that the manuscript improved in clarity and it is now more convincing that metabarcoding would be a reliable tool to determine trophic niche diversity.

I only found minor changes to incorporate:

- 1.106. 60-100m instead of 160-100m

- 1.269. *Pseudocalanus* in italics.

Decision letter (RSPB-2021-0908.R0)

14-May-2021

Dear Mr Novotny

I am pleased to inform you that your manuscript RSPB-2021-0908 entitled "DNA metabarcoding reveals trophic niche diversity of micro and mesozooplankton species" has been accepted for publication in Proceedings B.

The referees have recommended publication, but one reviewer also suggests some (extremely) minor revisions to your manuscript. Therefore, I invite you to respond to the referee's comments and revise your manuscript. Because the schedule for publication is very tight, it is a condition of publication that you submit the revised version of your manuscript within 7 days. If you do not think you will be able to meet this date please let us know.

1) A text file of the manuscript (doc, txt, rtf or tex), including the references, tables (including captions) and figure captions. Please remove any tracked changes from the text before submission. PDF files are not an accepted format for the "Main Document".

2) A separate electronic file of each figure (tiff, EPS or print-quality PDF preferred). The format should be produced directly from original creation package, or original software format. PowerPoint files are not accepted.

3) Electronic supplementary material: this should be contained in a separate file and where possible, all ESM should be combined into a single file. All supplementary materials accompanying an accepted article will be treated as in their final form. They will be published alongside the paper on the journal website and posted on the online figshare repository. Files on figshare will be made available approximately one week before the accompanying article so that the supplementary material can be attributed a unique DOI.

Sincerely,
Professor Hans Heesterbeek

Associate Editor

Comments to Author:

I'm pleased to inform you that all the reviewers have provided favorable feedback on your manuscript. Reviewer 2 though points out a few minor revisions that need to be addressed.

Reviewer(s)' Comments to Author:

Referee: 1

Comments to the Author(s).

Dear authors,

I am happy with the revised version of the manuscript, thank you for implementing the suggested corrections. I reviewed your manuscript with great pleasure and I would like to thank you for bringing new insights on plankton dynamics with your study.

Referee: 3

Comments to the Author(s).

First, I reviewed the responses provided regarding the comments. The authors replied thoroughly to all concerns, and included them in the manuscript.

Then, I reviewed the new version of the manuscript (i.e. without the track changes). I found that the manuscript improved in clarity and it is now more convincing that metabarcoding would be a reliable tool to determine trophic niche diversity.

I only found minor changes to incorporate:

- l.106. 60-100m instead of 160-100m

- l.269. *Pseudocalanus* in italics.

Referee: 2

Comments to the Author(s).

The authors have addressed all my concerns

Author's Response to Decision Letter for (RSPB-2021-0908.R0)

See Appendix B.

Decision letter (RSPB-2021-0908.R1)

17-May-2021

Dear Mr Novotny

I am pleased to inform you that your manuscript entitled "DNA metabarcoding reveals trophic niche diversity of micro and mesozooplankton species" has been accepted for publication in Proceedings B.

Your article has been estimated as being 9 pages long. Our Production Office will be able to confirm the exact length at proof stage.

Data Accessibility section

Open Access

Paper charges

Sincerely,

Proceedings B

Appendix A

Dear Professor Heesterbeeck,

We are pleased to see that our manuscript was well received by the editorial board and the three reviewers and are happy to submit a new version of the manuscript. We appreciate the detailed and constructive feedback from the reviewers, which helped us to improve our study.

We have changed the manuscript accordingly and implemented the comments in the revised version. Below you find a detailed reply to each comment, which is also highlighted as track changes in the revised version of the manuscript.

Thank you for continuously considering our manuscript for publication.

Best wishes

Reviewer(s)' Comments to Author:

Referee: 1

Comments to the Author(s)

The authors developed a metabarcoding analysis to explore the seasonal trophic dynamics among plankton organisms in a coastal environment of the Baltic Sea. The study is time-needed and provides important insights on species trophic niches, information that be applied for other similar ecosystems as well. The title of the study is informative and suitable, and the authors provide a well-written summary of their work in the abstract. The manuscript is coherent with comprehensive figures. Overall, I propose minor corrections which in my opinion will improve the manuscripts' readability and highlight the significance of this study. Please find below my comments and suggested corrections.

Major suggestions:

1. Lines 47-48: Please rephrase this sentence. Species are multifunctional and therefore can fit in different groups depending on their traits (e.g., in lines 48-53 the authors provide examples of how different taxonomic species can be grouped together based on their feeding traits). As authors mention in lines 65-66, is the limited quantified and qualified information that causes the misrepresentation of the diverse trophic niche and not the approaches per se.

We agree with the reviewer and removed this sentence.

2. To improve readability, move lines 55-64 after line 83 and link the sentences. For example: “[...] can be investigated simultaneously [35]. Metabarcoding can therefore provide significant insights on trophic interactions and better link the trophic niche diversity with energy flow. This can be of particular importance for coastal ecosystems with high ecosystem services that experience an increase in pico and filamentous cyanobacteria due to climate warming and eutrophication [21–23]. [...]”

We have changed this as suggested. See lines 74-83.

3. Move lines 83-85 to the discussion and expand with one sentence saying what is the main reason/limitation why there are only a few studies.

This sentence has been moved to a new context in the discussion. See lines 310-324.

4. Lines 87-90: This sentence is not well fitted with the intro. I propose to change added as a result output after line 93. “[...] Our results show that the trophic niche diversity extends beyond broad phylogenetic groups and size classes, and that rarely studied microzooplankton (rotifers and ciliates) fill an important role in the pathways of the coastal pelagic primary production.”

This suggestion has been implemented. See lines 89-91.

5. Sample region: Please add one sentence about why this location is chosen and the ecosystem properties of the sampling region (e.g., nutrient). Also please include a map in the Appendix.

We have added the suggested sentence on lines 94-98. In appendix 1, we have added fig. S1, a map that shows the station location.

6. Lines 270-272: This is a very interesting statement. Please provide more information with the relevant key message from the study [35] and how salinity is a potential driver.

This was a misunderstanding of the sentence as the cited study does not consider salinity as a potential driver. We realize that this sentence was misleading, and it has been changed. See lines 271-274.

7. Lines 289-292: This sentence is confusing as it includes copepods and protozoans. If you want to make this sentence copepods specific please remove the “our study shows that primarily smaller zooplankton, such as *Helicostomella* and *Keratella*” otherwise rephrase the whole sentence.

We have improved the sentence. See lines 290-294.

8. Lines 315-318, 325-329: Please move those statements in the discussion and expand further. In lines 325-329, the authors made an excellent point of how our modelling projections on energy flow would benefit by including trophic niches. But, as it has been mentioned in lines 315-318, metabarcoding cannot offer qualified data useful for models and data (e.g., stoichiometry, biomass, rates) from other approaches need to be combined. It would be helpful for the community if the authors can be more specific on the main limitations of their approach and which kind of data are necessary for implementing their data into models and which model design they think that are most suitable.

We are currently developing an approach for ecosystem modeling based on data from metabarcoding. Zooplankton diet data from metabarcoding is used to calculate a prey selectivity index between zooplankton and phytoplankton (See for instance [https://doi.org/10.1016/0304-3800\(92\)90016-8](https://doi.org/10.1016/0304-3800(92)90016-8)). This is later scaled with population biomasses (from monitoring) and metabolic energy demand to model fluxes between phytoplankton and zooplankton. While the full explanation of such model implementation lies outside the scope of this study, we agree with the reviewer that these statements should be clarified.

We have clarified our view on how DNA metabarcoding could be implemented in food web modeling on lines 316-324 and lines 334-335.

Minor:

1. lines 31-33: Please rephrase the first part of the sentence as is identical to the first sentence of the abstract.

The first paragraph of the abstract has been changed. See lines 13-14.

2. lines 45-47: This sentence needs to be rephrased. “True” is a strong word in the “true ecosystem structure”, please remove it. Additionally, like every theory, the trait-theory has its strengths and limitations, therefore the research question and model design will define if a trait-based approach is the best option.

We have changed this. See lines 44-46.

3. line 92: change to “a coastal region of the Baltic Sea”.

In a global context, the entire Baltic Sea is a coastal sea. However, from a Baltic perspective, Landsort Deep is considered a pelagic offshore station. We have changed this to “an offshore station in the Baltic Sea” at lines 87-88.

4. replace articles with studies throughout the discussion.

This has been changed throughout the manuscript.

5. Please include the DOI for all references and write in italics the name of species.

DOIs have been added for all references that are associated with a DOI.

Referee: 2

Comments to the Author(s)

The manuscript works on the diet preference to micro and mesozooplankton in the Baltic. The paper is clear and well written, with a few exceptions (indicated below).

There are some changes or additional information that, if provided, would improve the quality of the manuscript. I have also some statistical concerns on the ANOVA, since the authors are analyzing the same data twice. Also about the adequacy of ANOVA for this analyses.

We are grateful for these comments and agree that ANOVA on ranks was not the best approach for this data and have changed the analysis. See detailed answers to your comments below.

In Methods: Why the ciliates were sampled only from the top 10m? Wouldn't this make the study don't match each part? Could this impact how the results from Cilliates are interpreted, since they are only pulled from the surface.

Helicostomella is chosen as representative for the protozoan community, partially to illustrate that the method is not limited to metazooplankton. This role of *Helicostomella* has been clarified at lines 119-120 in the methods and 312-314 in the discussion. The sampling depth of the zooplankton samples, including *Helicostomella*, is now made more transparent in table S1 and revised fig. 3a.

For more details, see our answers to the depth-related comments below.

Line 143: BCI is then Bray-Curtis similarity? I suppose metaMDS calculated BC dissimilarity index (that I think that was the original definition, as dissimilarity, ranging from 0 for identical, to 1 with all non-overlapping) and then the authors transformed it into the BC similarity index (1-dissimilarity) and multiplied by 100? I am truly asking, since I have not seen this BCI before, and I think is just BC similarity (that the authors might refer then better as: “We used Bray-Curtis similarity index (in percentage) as an estimate to assess diet overlap between samples”). I am not sure why the multiplying by 100, instead of leaving is as is. I see this part potentially confusing for the reader without expertise in this kind of indices.

We agree that it is clearer to simply say “Bray-Curtis similarity”. This has now been changed. We have also removed the multiplication with 100. Changes have been made in the methods (line 145-146), throughout the results, and in fig. S4 for consistency.

Line 144: “Differences in diet overlap and differences in the proportion of the specific diet of consumers” but these two are the same: diet overlap is Bray-Curtis similarity index derived from the proportion of taxa. I see how the authors used them, but just to be aware that they are the same analyses based in the same data. These analyses are carried out with an ANOVA by ranks, that I do not think it solves the problem of the fact that the data the authors are analyzing are percentages. Maybe a PERMANOVA would be more adequate for the BC index, and chi-square with the relative abundances?

We understand the reviewer's point that it is difficult to follow the different tests, and how they differ. We have adapted the analysis according to the suggestions. Comparisons on Bray-Curtis differences are now based on PERMANOVA, and the Kruscal-Wallis analysis of distances has been removed.

We also understand the concern about using Kruscal-Wallis for the relative abundances. We are hesitant, however, to use the Chi-square test since total read abundance (n) has little biological relevance and is not equivalent to count data. We believe that beta regression is a better choice for proportions that are not based on frequencies (See for instance <https://doi.org/10.1111/2041-210X.13234>).

We have modified the description of statistical analysis in the method section (line 147) and calculated new statistics throughout the result section.

There are some other things that would make this paper more solid:

1) Could the authors provide a vertical profile plots of the Chl-a, temp and salinity of the water column? This would help understanding seasonality changes.

A new figure has been added to fig. S2a, containing Chl-a, salinity, and temperature profiles for the three sampling dates. Additionally, the new fig. S2b shows differences between zooplankton composition at the two different depth intervals: 0-30 and 30-60.

2) Is there a possibility that the different rotifer species show vertical partition, with then the associated changes in diet (that is suggested for copepods)? Again, without an explanation of the water column might be tricky. And since all net samples are mixed later on, it might be difficult to interpret, but it should be at least discussed.

We have inserted a new table (table S1) that clarifies what species were sorted from what depth sample. The consequences of this are included in fig. 3a, and the discussion.

Indeed, depth seems to have an important effect on the diet of the zooplankton. While *Pseudocalanus*, *Temora*, and *Centropages* came from below 30 meters, *Acartia*, rotifers, and cladocerans were sampled from the upper 30 m. Unfortunately, we do not have a higher resolution than these 30 m intervals. Also, the monitoring data is limited to these depth intervals – but it is definitely likely that differences between rotifers are due to differences in vertical distribution. Although this is an interesting research question, we think that a deeper discussion of depth and prey selectivity lies outside the scope of this study.

Concerning copepods, we have tempered lines 266-275 in the discussion.

3) Did the authors mixed also the two depths of the zooplankton? Are there differences in community between both regarding both the rotifers and the copepods? Any information from this would also help discussing the differences between rotifers and copepods.

The zooplankton samples were not mixed between depth layers as the depths were treated separately. This was not clearly described in the previous version and we have corrected this in the sampling method description (lines 101-109) and inserted a new table (table S1) that clarifies from which depth layers the species were sorted. The consequences of this are included in fig. 3a, and the discussion (lines 266-271). Also, a figure about zooplankton distribution in 0-30 and 30-60 m fractions are included along with the vertical profiles in fig. S2.

Minor:

Line 140: maybe better using “heterogeneous” instead of “heterogenic” (this is a term for allele studies).

This has been corrected. See line 142.

Referee: 3

Comments to the Author(s)

Phylogeny and body size are the traditional theories to investigate trophic interactions and predict food web dynamics and ecosystem functioning. In this work, Novotny, Zamora-Terol and Winder highlight the benefit of using DNA metabarcoding to investigate trophic interaction in a pelagic food web in the Baltic Sea, focusing on the functional diversity. Indeed DNA metabarcoding can describe the functional diversity in the food web. With this method, the authors were able to identify alternative pathways of energy, which stabilize the pelagic food web. As such, the topic and discussion brought by this article is very valuable. The paper is well written, and I found minor points to be unclear:

- The authors bring a valuable point towards the interesting functions of rotifers and ciliates in trophic niche diversity, which conflicts with the main theories of trophic interaction based on phylum and size (l. 232-235). This conclusion is based on “selected zooplankton” with one species of ciliates and three species of rotifers, which -I believe- was dependent on the abundance of these organisms. The manuscript would benefit with additional information about how this method could be relevant for rarer species.

In this study, we have aimed to cover both species with the highest abundance as we assume that they have the largest significance for ecosystem dynamics, but also species of a broad taxonomic range. As examples, we have included the ciliate *Helicostomella* and *Synchaeta baltica* in August to illustrate that the method is not limited to metazoans and species with low abundance, respectively (see fig. 1).

Individuals of abundant zooplankton were selected from depths where they were abundant. We have clarified this on lines 111-112, table S1 and fig. S2.

Any limitations in applying this method to rare taxa, would lie in the sampling method, and not in the metabarcoding. We have included a sentence about rarer species in the discussion. See lines 312-316.

- In the introduction (l. 65-69), the authors argued that metabarcoding would be more efficient than experiments (e.g. dilution techniques). However, the authors concluded that this method “combined with validating experiments” would be suitable method for food web modeling and ecosystem networking analysis. The conclusion would be more informative with an example of such validating experiment.

We have tempered the discussion and clarified our view on how DNA metabarcoding could be implemented in food web modeling at line 316-324 and at line 334-335.

- In the results section. According to the statistical guidelines, the authors should add the degree of freedom of the Kruskal-Wallis test; and use the lowercase p for the p.value.

This has been changed. On the suggestion of reviewer 2, Kruskal-Wallis tests have been replaced by Beta-regression. Degrees of freedom are now included in the statistical results.

- L.101. I understood that water samples served as control; it is better to state it clearly.

Water samples were used partly as a control, but also to identify available prey organisms. We have now clarified this on line 101.

- L.103. The water was filtered “sequentially” from the smaller to larger pore size. Should not be the reverse order?

Yes indeed, the water first passed through the filter with the larger pore size. The error is now corrected on lines 104-105.

- L.160-161. I do not understand how the number of RSVs in the bulk water samples (2771) and in the selective zooplankton samples (1799) can be larger than the total (1483).

Thank you for noticing this. These numbers were unfortunately based on wrong calculations. The values (both for *I6S* and *I8S*) have been checked and corrected. See lines 159-162.

- L.166. I believe it is better to be consistent between the text and the figures. Here, in the text, the authors refer to 90% of heterotrophic bacteria; while in Figure S2 they refer to 85% of Proteobacteria.

We agree, and made the text and figure consistent and report the percentage on “Proteobacteria” on line 166.

- L.176. I was confused with the sudden division into spring vs summer because there are three dates of observations. Then, I understood that the results were referring to the first observation (i.e., march). Consider replacing “In spring” by “In March”.

We now have clarified the transition between March and June throughout the results. See lines 174-198.

- L.182. Figure 3a instead of figure 4?

Some figures have changed in the new version. All figure references have been adjusted.

- L.192. The decline of *Peridiniella* is not shown in Figure 1. Please, mention the phylum corresponding to *Peridiniella*.

This has been adjusted. See line 191.

- L.206. For consistency, please refer to cyanobiaceae (?)

This has been adjusted. See line 205.

- L.260. There is a problem with the reference Arndt (1993)

The reference has been fixed. See lines 258-259.

- L.312-314. Few elements reduced the clarity of the key message. Please, consider removing “rather than focusing on the sequence read abundance of prey” because the relative comparison is using the read abundance; and “without the assumption of a relationship between read-count and biomass” as the authors state that they do not have such information (l.317).

We have improved this. See line 318-319.

- Figure 1a. There are 8 dynamics shown and 7 with the legends.

The legend of *Centropages* was missing but is now included in fig. 1a.

- Figure 3. To link the panels a and b, I suggest to have the same numbered labels.

The numbered legend has been merged, and the two panels are now easier to compare. See fig. 3.

- Figure 4. Information about the size of the organisms is only mention in the discussion. As the authors aimed at showing differences between phylum/size and metabarcoding method, the authors could mention the size in this (nice!) figure 4a summarizing the results/discussion.

This is a good suggestion. Body size is now included in fig. 4a.

Appendix B

Dear Professor Heesterbeeck,

Thank you for accepting our manuscript for publication in Proceedings B. We are pleased to see that our reviewed manuscript has been well received and we are happy to submit a new version of the manuscript. We appreciate all work from editors and reviewers during the revision process.

Attached is a version with the reviewer's comment implemented along some minor additional language edits.

Best wishes
The authors

Associate Editor

Comments to Author:

I'm pleased to inform you that all the reviewers have provided favorable feedback on your manuscript. Reviewer 2 though points out a few minor revisions that need to be addressed.

Reviewer(s)' Comments to Author:

Referee: 1

Comments to the Author(s).

Dear authors,

I am happy with the revised version of the manuscript, thank you for implementing the suggested corrections. I reviewed your manuscript with great pleasure and I would like to thank you for bringing new insights on plankton dynamics with your study.

Referee: 3

Comments to the Author(s).

First, I reviewed the responses provided regarding the comments. The authors replied thoroughly to all concerns, and included them in the manuscript.

Then, I reviewed the new version of the manuscript (i.e. without the track changes). I found that the manuscript improved in clarity and it is now more convincing that metabarcoding would be a reliable tool to determine trophic niche diversity.

I only found minor changes to incorporate:

- l.106. 60-100m instead of 160-100m
- l.269. *Pseudocalanus* in italics.

Thank you for noticing these mistakes. The changes are incorporated.

Referee: 2

Comments to the Author(s).

The authors have addressed all my concerns